# Effects of Traumatic Brain Injury on the Gut Microbiota Composition and Serum Amino Acid Profile in Rats

**DOI:** 10.3390/cells11091409

**Published:** 2022-04-21

**Authors:** Anastasiia Taraskina, Olga Ignatyeva, Darya Lisovaya, Mikhail Ivanov, Lyudmila Ivanova, Viktoriya Golovicheva, Galina Baydakova, Denis Silachev, Vasiliy Popkov, Tatyana Ivanets, Daria Kashtanova, Vladimir Yudin, Valentin Makarov, Ivan Abramov, Mariya Lukashina, Vera Rakova, Anzhelika Zagainova, Dmitry Zorov, Egor Plotnikov, Gennadiy Sukhikh, Sergey Yudin

**Affiliations:** 1Federal State Budgetary Institution “Centre for Strategic Planning and Management of Biomedical Health Risks” of the Federal Medical Biological Agency, 123182 Moscow, Russia; ataraskina@cspfmba.ru (A.T.); dlisovaya@cspfmba.ru (D.L.); mivanov@cspfmba.ru (M.I.); lgivanova@cspmz.ru (L.I.); dr.kashtanova@gmail.com (D.K.); vyudin@cspfmba.ru (V.Y.); makarov@cspfmba.ru (V.M.); iaabramov@cspmz.ru (I.A.); mlukashina@cspmz.ru (M.L.); rakova@cspmz.ru (V.R.); azagaynova@cspmz.ru (A.Z.); yudin@cspmz.ru (S.Y.); 2A.N. Belozersky Institute of Physico-Chemical Biology, Lomonosov Moscow State University, 119992 Moscow, Russia; viktoriia.golovicheva@yandex.ru (V.G.); silachevdn@belozersky.msu.ru (D.S.); popkov.vas@gmail.com (V.P.); zorov@belozersky.msu.ru (D.Z.); plotnikov@belozersky.msu.ru (E.P.); 3Research Center for Medical Genetics, Moscow 115522, Russia; gb2003@yandex.ru; 4V.I. Kulakov National Medical Research Center of Obstetrics, Gynecology and Perinatology, 117997 Moscow, Russia; t_ivanets@oparina4.ru (T.I.); g_sukhikh@oparina4.ru (G.S.)

**Keywords:** gut microbiota, gut–brain axis, traumatic brain injury, amino acids, complications of traumatic brain injury

## Abstract

Traumatic brain injury (TBI) heavily impacts the body: it damages the brain tissue and the peripheral nervous system and shifts homeostasis in many types of tissue. An acute brain injury compromises the “brain–gut-microbiome axis”, a well-balanced network formed by the brain, gastrointestinal tract, and gut microbiome, which has a complex effect: damage to the brain alters the composition of the microbiome; the altered microbiome affects TBI severity, neuroplasticity, and metabolic pathways through various bacterial metabolites. We modeled TBI in rats. Using a bioinformatics approach, we sought to identify correlations between the gut microbiome composition, TBI severity, the rate of neurological function recovery, and blood metabolome. We found that the TBI caused changes in the abundance of 26 bacterial genera. The most dramatic change was observed in the abundance of *Agathobacter* species. The TBI also altered concentrations of several metabolites, specifically citrulline and tryptophan. We found no significant correlations between TBI severity and the pre-existing gut microbiota composition or blood metabolites. However, we discovered some differences between the two groups of subjects that showed high and low rates of neurological function recovery, respectively. The present study highlights the role of the brain–gut-microbiome axis in TBI.

## 1. Introduction

Globally, traumatic brain injury (TBI) remains a leading cause of death and disability and presents an urgent challenge for healthcare systems [1] It is known to cause massive lesions in the central nervous system (CNS), gastrointestinal tract (GI), and respiratory- and cardiovascular systems [2] Recently, profound physiological effects of TBI on the GI tract, its functioning, and its microbiome [3] have been widely discussed. Evidence suggests that post-TBI GI dysfunctions can exacerbate brain damage and trigger a chain reaction: each component of the ‘gut–brain axis’ (GBA), the bidirectional signaling system that links the GI tract and the CNS, also referred to as the ‘brain–gut-microbiome axis’ to emphasize the role of commensal microorganisms [4], affects the other [5]. This impedes proper functioning of the GBA and can potentially lead to serious changes in the body and cause diseases of the CNS and GI tract [6,7].

For instance, patients with CNS injuries are known to suffer from severe GI dysfunctions, most commonly lower GI motility and gastric mucosal damage. There are several probable underlying mechanisms of post-TBI GI dysfunctions: loss of enteric neurons and changes in their structure with subsequent weakening of the intestinal peristalsis [8,9]; and intestinal barrier disruption. A CNS injury leads to a massive release of hormones, such as adrenocorticotropic hormone, resulting in gastric mucosal damage [10]. Furthermore, sympathetic and parasympathetic activation produces neurotransmitters that can indirectly affect the mucosal barrier and compromise mucosal integrity.

A disrupted GI barrier is closely associated with changes in the microbiota composition and its functioning [11,12]. Multiple studies have demonstrated in both animals [13,14,15] and humans [16,17] that CNS injuries significantly affect the abundance and diversity of bacterial species that inhabit the intestine. The commensal microbiota appears to contribute to strong GI and blood–brain [18,19] barriers and promote mucus production and intestinal epithelial-cell regeneration. Moreover, the GI microbiome is likely to aid maturation and functioning of microglia [2,20], a potential mediator between the commensal microbiota and CNS. Commensal bacteria in the GI also produce many metabolites, such as short-chain fatty acids (SCFAs). Through monocarboxylate transporters located on endothelial cells, these molecules can exit the gut, cross the blood–brain barrier, and directly or indirectly modulate the CNS functions, including emotions and cognition, and even lead to pathological conditions [21]. Some bacterial species in the GI tract can synthesize chemicals that act as neurotransmitters and neural regulators and affect neuronal signaling [22].

The gut microbiota can also activate pattern-recognition receptors (PRRs) on GI immune cells and trigger abnormal secretion of inflammatory cytokines [23], causing immune dysfunctions not only in the GI tract but also in other organ systems.

Impaired functioning of the intestinal barrier may lead to the release of bacteria or their metabolites into the bloodstream and cause local or systemic inflammation [24]. Thus, changes in the GI environment caused by CNS injuries usually trigger dysbiosis that might result in systemic inflammation and poorer neurological prognosis. A better understanding of post-TBI changes in the gut microbiota composition can reveal mechanisms of gut–brain interactions and contribute to improved management of the CNS dysfunctions, including TBI.

The study aims to evaluate post-TBI changes in the gut microbiome composition and metabolic profile, including amino acids and acylcarnitines, and assess the effect of the pre-injury microbiome on TBI outcomes.

## 2. Materials and Methods

### 2.1. Rats

We modeled traumatic brain injury (TBI) in 25 3–4-month-old male rats of the Wistar outbred strain, weighing 350–400 g each. The rats had unlimited access to food and water and resided in temperature-controlled cages (20 ± 2 °C) lit from 9:00 a.m. to 9:00 p.m. The study followed the guidelines of the Federation of Laboratory Animal Science Associations (FELASA) and was approved by the Animal Ethics Committee of the Belozersky Institute (Protocol 9/20 from 2 September 2020).

### 2.2. Traumatic-Brain-Injury Model

The rats were anesthetized with i/p injections of 300 mg/kg (6%) chloral hydrate for all procedures. To ensure proper pre-and postoperative pain relief, bupivacaine ointment, a topical long-acting local anesthetic, was applied repeatedly. During the intervention and until awake, the animals were heated with a feedback-controlled heating pad supplemented with an infrared lamp to maintain their core temperature at 37.0 ± 0.5 °C.

In this study, we used a modified model of focal open severe brain trauma in rats [25,26]. To induce trauma, the rats were positioned in a stereotaxic frame (NeuroStar Robot Stereotaxic, Germany); their scalps were disinfected with Ioprep solution; aseptic techniques were used throughout the intervention. Part of the scalp was removed and trepanation was performed on the right frontal part of the skull above the sensorimotor cortex zone (the trepanation window). An unfixed Teflon footplate, 4 mm in diameter with depth of insertion of 2.5 mm, was positioned in way that would induce trauma upon impact. The impact was exerted with a 50 g load positioned at the height of 10 cm, which was let loose along a directing rail. To localize the sensorimotor cortex zone, the following stereotaxic coordinates were used: +4 to −3 mm anterior and posterior from bregma and +1 to +4.5 mm lateral from the midline. The incisions were then stitched with Ethicon threads (Size 3–0).

The damage was quantified based on MRIs, obtained seven days after the TBI. Neurological status was determined using the Limb Placement Test before the TBI (at the baseline, day 0) and on days 1, 3, and 7 after the TBI. The Cylinder Test was used to assess forelimb asymmetry seven days after the TBI.

### 2.3. Magnetic Resonance-Imaging Studies of the Brain Damage

The brain damage was quantified based on MRI, obtained 7 days after the intervention [27] with a 7T MRI scanner (Bruker BioSpec 70/30 USR; Bruker BioSpin, Ettlingen, Germany) using 86 mm volume-type RF transmit resonator and a phased array rat head surface coil. Before scanning, the animals were anesthetized with 2–2.5% isoflurane and a mixture of oxygen and air. The rats were placed in the prone position on a water-heated bed with their heads immobilized with snout masks and masking tapes. The imaging protocol included a T2-weighted sequence (repetition time = 4500 ms; echo time = 12 ms; slice thickness = 0.8 mm).

### 2.4. Neurological Deficit Assessment

#### 2.4.1. Limb Placement Test

A seven-step modification of the Limb Placement test (LPT) was used to assess forelimb and hindlimb responses to tactile and proprioceptive stimulation [28,29]. The rats had been habituated to handling for three days before the test. For each task, the following scores were used: 2 points for normal response; 1 point for delayed and/or incomplete response; 0 points for no response. The mean score was calculated across all tasks. All limbs were tested. In the first task, the rat was held by a researcher with its forelimbs positioned on the edge of the table. Each forelimb was gently pushed off the edge one after the other and the response of the animal was registered. Normally, the rat immediately placed its limb to its original position. In the second task, the first task was repeated with the animal’s head raised at 45 degrees so that it could not see the surface of the table or contact it with its vibrissae. In the third and fourth tasks, the rat was placed along the edge of the table and its fore- and hindlimbs were pushed aside. In the fifth test, the rat was placed on its hindlimbs on the edge of the table and then each limb was sequentially pushed down. In the sixth task, the rat was placed on its forelimbs near the edge of the table and gently pushed ahead towards the edge. Healthy rats resisted with their limbs, whereas rats with cerebral ischemia did not resist and their limbs slipped off the edge of the table. In the seventh task, the rat was held at the base its tail and slowly lowered to the surface of the table. Normally, the rat stretched its forelimbs forward in approximately 10 cm to the surface of the table.

#### 2.4.2. Cylinder Test

The Cylinder Test, or spontaneous exploration of the cylinder walls [30], was used to assess asymmetry in forelimb use: the movements of the rats in a transparent cylinder (30 cm in height and 20 cm in diameter) were recorded for 5–8 min using a camcorder positioned above the cylinder. Independent uses of the contra- and ipsilateral forelimbs while standing on the hind limbs and their simultaneous (combined) use were counted with the following formula: (contr + 1/2 × simult)/(ipsi + simult + contr) × 100, where contr and ipsi denote the use of contralateral (damaged) and ipsilateral limbs, and simult denotes simultaneous use of both forelimbs. Forelimb-use asymmetry was assessed on day 7 after the TBI.

#### 2.4.3. Assessment of Neurological-Deficit Dynamics

We divided the post-TBI rats into two groups based on neurological-function-recovery dynamics, which were assessed on days 1, 3, and 7 using the LPT data. Group 1 included the rats that showed a positive dynamic, i.e., had higher LPT scores on day 7 vs. 1. Group 2 consisted of the rats showing no positive dynamic by day 7.

### 2.5. Sampling

Blood samples were collected from the jugular vein before and on days 3 and 7 after the TBI to determine the concentrations of amino acids and acylcarnitines in the blood serum. Fecal samples were collected before and on day 7 after the TBI.

### 2.6. Tandem Mass Spectrometry

We analyzed amino acid and acylcarnitine levels in blood serum before and on days 3 and 7 after the TBI by FIA–MS/MS using the NeoGram Amino Acids and Acylcarnitines Tandem Mass Spectrometry Kit (Perkin Elmer Life and Analytical Sciences, Waltham, MA, USA), Sciex QTrap 3200 quadrupole tandem mass spectrometer (Sciex, Framingham, MA, USA) operating in the positive-ion electrospray ionization mode and Shimadzu 20LC system (Shimadzu, Kyoto, Japan).

Amino acids and acylcarnitines were extracted from 5 µL of plasma with a methanol/water (75:25) solution containing the stable isotope-labeled internal standards. The samples were diluted in butanolic HCl, dried at 60 °C, and reconstituted with acetonitrile/water (80:20) solution containing acetic acid. A 20 µL aliquot of the sample was directly injected into the MS/MS system.

Analyte concentrations were measured by comparing the responses to each amino acid and acylcarnitine and responses to the corresponding stable isotope-labeled internal standards. Calculations were performed on ChemoView software version 2.0.2 (Sciex, Framingham, MA, USA).

### 2.7. Assessment of Serum Markers of Brain Injury

Serum levels of neuron-specific enolase (NSE) in the samples collected on days 0 (n = 26) and 3 (*n* = 26) were measured using the Elecsys Electrochemiluminescent Immunoassay NSE^®^ (COBAS; Roche Diagnostics, Rotkreuz, Switzerland). The measurements ranged from 0.1 ng/mLto 300 ng/mL.

### 2.8. 16S rRNA Sequencing (V3–V4 Regions)

#### 2.8.1. Fecal DNA Extraction and Amplification

Fecal DNA was extracted using the QIAamp Fast DNA Stool Mini Kit according to the manufacturer’s instructions.

Before the PCR, the DNA was diluted 500 times. One-round PCR was used to amplify the V4-region of the 16S rRNA gene using the Veriti thermal cycler (Applied Biosystems, Foster, CA, USA) according to the following protocol:98 °C for 30 s;30 cycles: 98 °C for 15 s;      58 °C for 15 s;      72 °C for 15 s;      72 °C for 1 min;Storage at 4 °C.

#### 2.8.2. 16S rRNA Sequencing

The PCR products were purified using the Cleanup Mini kit for DNA extraction (Eurogen). The concentration of the DNA libraries was measured using the Quant-iT™ dsDNA High-Sensitivity Assay Kit with the Qubit^®^ fluorometer (Invitrogen, Carlsbad, CA, USA). The purified amplicons were mixed in an equimolar manner according to their concentrations. The quality of the sequence libraries was assessed by agarose gel electrophoresis.

Further sample preparation and sequencing were performed on the MiSeq platform (Illumina, San Diego, CA, USA) using the MiSeq reagent kit V2 (500 cycles) following the manufacturer’s instructions. Primary processing (extraction of barcodes) was conducted as described earlier [31].

### 2.9. Bioinformatic Analysis

The raw data were denoised using the Divisive Amplicon Denoising Algorithm (DADA2) v1.18.0 (available at https://github.com/benjjneb/dada2; accessed on 12 April 2021) with the microbiome analysis pipeline. Forward and reverse reads were truncated at 275 bp and 205 bp, respectively, by using read quality scores and the filterAndTrim function set at maxEE = c(2,2), truncQ = 2, snf minQ = 6. Singleton and chimeric sequences were removed by the DADA2 error model and removeBimeraDenovo function. The taxonomic classification (phylum to genus) was performed with the Silva Database (Silva) naïve Bayesian classifier (implemented in DADA2) and Silva nr v132 training set. To analyze alpha and beta diversities, the classified reads were reduced to the same number of reads (3000 reads per sample). Alpha diversity was assessed for each sample using the Shannon diversity index. Beta diversity was assessed using the microbial dissimilarities matrix (Bray–Curtis) obtained using the vegdist function from the vegan R-package. Nonmetric multidimensional scaling (NMDS) of microbial communities was performed using the vegan R package and Bray–Curtis dissimilarity values.

The R package MaAsLin2 (Microbiome Multivariable Associations with Linear Models) from MaAsLin2 framework was used to determine multivariable associations between relative taxonomic abundance and TBI.

### 2.10. Statistical Analysis

To describe the data, we calculated the means and standard deviations, or interquartile ranges (IQR). Due to abnormal distribution of the data (on the abundance of amino acids and microorganisms), we used nonparametric tests, including the Wilcoxon test and Mann–Whitney U test, to evaluate the differences between the groups and time-points (days 0, 3, and 7). To adjust the level of significance, we used the Bonferroni multiple-testing correction.

#### Principal Component Analysis (PCA)

To analyze amino acids and microbiota compositions, we reduced the dimensionality of large data sets by transforming them into a new set of variables using Principal Component Analysis (PCA).

To analyze associations between amino acid levels and the extent of brain injury, we used the ordinary least-squares linear-regression model from Statsmodels v0.12.2 (Python 3.8 package) module. We modeled the data with the following function:$$y=\beta_{1}x_{1}+\beta_{2}x_{2}+\beta_{0},$$
where *y* represents the target variable (binary variable representing the days after and before TBI, a group index denoting the post-TBI dynamics); $x_{i}$ represent the principal components. To find parameters, we minimized the sum of the square differences of the observed and expected values. To evaluate the significance of the variables, we used a *t*-test and calculated *p*-values. PCA was based on the beta-diversity metrics and was performed using the R package (PCAtools).

To compare the PCs for different timepoints, we used a generalized *t*-test for a 2-dimensional space (2*t*-test).

To estimate the differences between the pre- and post-TBI samples in a two-dimensional (PC1/PC2) space, we built a linear support-vector machine (SVM). The SVM algorithm creates a hyperplane that separates the data into two classes with the maximum margin. The SVM classification was then verified with a *t*-test based on the distances to the separating hyperplane.

## 3. Results

### 3.1. Post-TBI Changes in Neurological Signs and MR Scans

The results of the LPT revealed the development of a post-traumatic functional deficit in the left limbs (contralateral to the damaged cortex); the right limbs remained fully functional. Since there was a significant difference in the dynamics of neurological function recovery based on the LPT, for further analysis, the rats were divided into two groups: the rats that demonstrated increased LPT scores on day 7 and were considered to have a positive dynamic (Group 1); and the rats demonstrating no recovery of the sensorimotor functions on day 7 (Group 2) (Figure 1A).

Using the Cylinder Test, we found that the TBI resulted in the asymmetric use of the forelimbs. Normally, rats use both the left and right limbs equally when examining the walls of the glass cylinder. Damage to the sensorimotor area of the right hemisphere caused a two-fold decrease in the use of the contralateral limb (the left forelimb) on day 7 after the TBI: to 24.1 ± 6.9% in Group 1 and to 23.53 ± 3.8% in Group 2 (Figure 1B).

The TBI caused an extensive damage to the sensorimotor cortex, as shown by the MRIs obtained on day 7 after the TBI (Figure 1C). The MRIs revealed no significant difference in brain damage between the groups: 80.1 ± 30.86 mm^3^ in Group 1 and 88.1 ± 45.34 mm^3^ in Group 2 (Figure 1D).

### 3.2. Post-TBI Changes in Serum Markers of Brain Injury

We used NSE as a specific severity marker of the brain injury. We observed a significant increase in NSE levels in the post-TBI serum samples, compared to those collected at baseline (Figure 2A). There were no differences in the NSE levels between Groups 1 and 2 on day 0; however, on day 3, NSE was significantly higher in Group 1 (Figure 2B).

### 3.3. Post-TBI Changes in the Gut Microbiome Composition and Metabolic Profile

#### 3.3.1. Taxonomic Composition and Differential Abundance in Bacterial Communities

We filtered and trimmed the reads, clustered them into operational taxonomic units (OTUs) of 16S rRNA with a 97% similarity threshold, and removed OTUs with a relative abundance of less than 0.005%.

The most abundant phyla in the pre-TBI samples were *Firmicutes* (66.1%) and *Bacteroidetes* (22.7%). On day 7 after the TBI, the proportion of *Firmicutes* increased to 68.0% and the proportion of *Bacteroidetes* decreased to 19.8% (Figure 3). Species of the candidate superphyla *Patescibacteria* and *Campilobacterota* were also very abundant and present in most samples. Figure 4 shows the phylogenetic tree of the rats’ fecal microbiota.

We assessed changes in the gut microbiota composition before and on day 7 after the TBI using the MaAsLin. Figure 5 shows the significant changes detected with the MaAsLin. The TBI affected the abundance of numerous bacterial species: the number of 14 bacterial genera drastically reduced while the number of 12 bacterial genera increased. The most pronounced decrease was observed in the *Agathobacter* species, the genera *Faecalibacterium* and *Paraprevotella*, and *Eubacterium coprostanoligenes*. We detected the most significant post-TBI growth in two *Rikenellaceae* RC9 gut-group species, members of the *Prevotellaceae* family (including *Prevotella*), and *Lactobacillus*, *Turicibacter*, and *Helicobacter* species.

To measure the post-TBI changes in the gut microbiota, we analyzed the Bray–Curtis dissimilarity between the samples collected on days 0 and 7 using Principal Coordinate Analysis (PCA). Principal component 1 (PC1) explained 35.64% of the total variance and was positively correlated with several taxonomic groups, including the genera *Barnesiella*, *Alistipes*, *Faecalibacterium*, and *Akkermansia*. Principal component 2 (PC2) explained 14.21% of the total variance and was positively correlated with the genera *Escherichia/Shigella*, *Catenibacterium*, *Bifidobacterium*, and *Prevotella*.

We compared PC1 and PC2 for the pre-and post-TBI samples using the two-sample *T*-test and Kolmogorov–Smirnov test; both tests demonstrated significant differences (*p* = 0.007 and *p* = 0.005, respectively). To verify the quality of the discrimination between the pre-and post-TBI samples that was based on PC1 and PC2, we used a support-vector-machine (SVM) model: we plotted the data in PC1, PC2 plane and separated them with a hyperplane; the distance from each sample datum to the separating hyperplane was calculated and compared for days 0 and 7 using a *t*-test (Figure 6B). We found significant differences (*p* = 0.018), indicating that the gut microbiota composition had changed between the two analyzed time points.

The linear-regression analysis did not reveal any significant associations between the extent of brain injury and post-TBI gut microbiota composition.

#### 3.3.2. Analysis of Microbiota Diversity

To measure changes in α-diversity on days 0 and 7, we calculated the Shannon indices and compared them using a *t*-test. We found that the TBI caused a substantial decrease in α-diversity of the fecal microbiota seven days after the trauma (*p* = 0.03) (Figure 7).

Beta-diversity was measured based on the Bray–Curtis dissimilarity and visualized using nonmetric multidimensional scaling (NMDS). The pre-TBI and post-TBI samples demonstrated a significant difference in the microbiota composition (*p* = 0.01) (Figure 8).

#### 3.3.3. Changes in Serum Amino Acid Levels

We analyzed the changes in the amino acid levels using PCA. PC1 explained 48% of the total variance. The ratio of free carnitine and its derivatives (palmitoylcarnitine + stearoyl carnitine) was positively and strongly correlated with PC1 (0.97) while glutamic acid and alanine demonstrated a negative correlation (−0.18 and −0.14, respectively).

PC2 explained 28% of the total variance. Amino acid level correlations with PC2 were mostly positive (negative associations were weak and below 1%): the loading ratio for the free carnitine and its derivatives (palmitoylcarnitine + stearoyl carnitine) was 0.71; for proline, 0.57; for citrulline, 0.25; for free carnitine, 0.22.

To analyze differences between the pre-and post-TBI amino acid content in blood, we used a 2*t*-test (*p* < 0.00013). There was no significant change in the amino acid content before and on days 3 and 7 after the TBI (*p* = 0.378 and *p* = 0.003, respectively); the changes between days 7 and 3 were not significant either (*p* = 0.002) (Figure 9).

We used the Wilcoxon test method to compare amino acid levels in blood serum on days 3 and 7 after the TBI. Table 1 shows significant associations as the mean (IQR) and *p*-values (differences were considered significant from a *p*-value of <0.00064).

Figure 10A shows fold changes in amino acid levels before and on day 7 after the TBI. Significant changes are marked according to the Wilcoxon test results. Figure 10B shows fold changes in the levels of the amino acids before and on day 3 after the TBI.

Based on the linear-regression analysis, there were no significant associations between amino acid levels in blood serum and the extent of the brain injury (PC1 and PC2) *p*-values for PC1 and PC2 were 0.564 and 0.09, respectively, on day 3; and 0.109 and 0.275, respectively, on day 7.

### 3.4. Changes in the Gut Microbiome Composition and Metabolic Profile in Groups with Different Post-TBI Recovery Rates

#### 3.4.1. Taxonomic Composition and Differential Abundance of Bacterial Communities

Changes in the gut microbiota composition were assessed using the MaAsLin. There were no significant differences in the taxonomic composition of the pre- and post-TBI fecal microbiota samples between Groups 1 and 2, including the genera, the abundance of which changed significantly after the TBI in the entire sample, such as *Agathobacter*, the *Rikenellaceae* RC9 gut group, and *Prevotella*.

Figure 11 shows fold changes in the relative abundance of the gut microbiota after the TBI in two groups.

#### 3.4.2. Analysis of Microbiota Diversity

The differences in the Shannon indices between the Group 1 and Group 2 samples were not significant (*p* = 0.56) (Figure 12).

#### 3.4.3. Changes in Serum Amino Acid Levels in Group 1 and Group 2

The pre-TBI level of Proline in Group 2 was significantly higher (Table 2, Figure 13). The fold change of ratio phenylalanine/tyrosine in the post-TBI samples between days 3 and 7 was also significantly higher in Group 2 (Table 2, Figure 14).

## 4. Discussion

TBI leads to multiple acute changes in the human body, specifically in the GI tract and its commensal bacteria. Several studies have demonstrated that TBI causes rapidly developing changes in the gut microbiome [13,15,32,33]. Some researchers even reported alterations persisting up to 30 days after TBI [32]. In our study, rats subjected to TBI experienced a significant gut microbiota depletion, compared to the baseline: the Shannon indices were higher for the pre-TBI microbiome samples. Diverse gut microbiota is known to be crucial to the proper functioning of the GI tract, retaining health, and preventing diseases. We analyzed alpha-diversity in the gut microbiota in pre-and post-TBI rats. There was no association between the baseline diversity and the neurological recovery dynamics; there was no difference in the Shannon indices between the rats demonstrating a negative neurological dynamic and those showing a positive neurological dynamic. A probable explanation could be that all animals were subjected to relatively similar impacts to induce TBI, which resulted in minimal differences in the severity of brain damage and neurological dynamics. Our findings on alpha-diversity are consistent with the results of other studies on the post-TBI gut microbiome in animal models [14,15].

However, we observed significant changes in the post-TBI abundance of some bacterial taxa. The TBI caused a shift in the proportion of 26 genera: the abundance of two *Agathobacter* species, Gram-positive anaerobic butyrate producers frequently found in animal and human feces, decreased dramatically. Butyrate-producing microorganisms are essential for the host: butyrate is the main energy source for colonic mucosa cells; furthermore, it is involved in regulating genes, inhibiting inflammation, and inducing differentiation and apoptosis of the host cells [34]. The depletion of *Agathobacter* species probably affected the metabolism both in the host (including amino acid metabolism) and its commensal microbiota. Notably, *Agathobacter* has been previously shown to be downregulated in patients with mild TBI [34,35]. Moreover, Hua et al. reported lower levels of *Agathobacter* in patients with sleep disorders [36], indicating that these microorganisms could be involved in CNS-related disorders.

We also found that *Faecalibacterium* and *Eubacterium*, beneficial butyrate-producing bacteria, were also sensitive to the brain injury: their relative abundance decreased drastically after the TBI. These results are consistent with the findings reached by other authors. For instance, Soriano et al. documented a decrease in *Eubacterium* abundance in athletes exposed to subconcussive impacts [17]. Bannerman et al. reported that a decreased abundance of *Faecalibacterium* was associated with exacerbated pain in patients with myalgic encephalomyelitis and chronic fatigue syndrome [37]. Interestingly, lower levels of *Faecalibacterium* correlated with sleep problems and manifestation of autism spectrum disorder [36].

The number of several bacterial species grew after the brain injury: there was a significant increase in the relative abundance of species belonging to the families *Prevotellaceae* and *Rikenellaceae*. The majority of *Prevotella* species are considered commensals because of their extensive presence in healthy individuals. However, some *Prevotella* strains are believed to be opportunistic pathogens since they have been isolated from bacterial abscesses in various parts of the body [38]. Some authors describe the proinflammatory role of *Prevotella* in certain diseases, such as rheumatoid arthritis [39] and periodontitis [40]. Fewer publications describe the role of *Rikenellaceae* in health and disease. Vogt et al. found that patients with Alzheimer’s disease were more likely to have a higher abundance of *Rikenellaceae* than healthy individuals [41]. Not surprisingly, in our study, the abundance of *Helicobacter*, that is more likely to be an opportunistic or latent pathogen, also increased significantly, compared to the baseline. Similar findings were reported by Matharu et al. in rats after repeated mild traumatic brain injury [32].

Some of these changes are similar to those observed in clinical studies of the intestinal microbiota in TBI patients. Analysis of healthy volunteers and patients 24 h after the TBI event demonstrated that the abundance of *Enterococcus*, *Parabacteroides*, *Akkermansia*, and *Lachnoclostridium* significantly increased, while that of *Bifidobacterium* and *Faecalibacterium* decreased in TBI patients [42]. Studies in patients with ischemic stroke also demonstrated significant gut microbiota dysbiosis with an increased abundance of short-chain fatty-acid (SCFA)-producing bacteria, such as *Odoribacter* and *Akkermansia*, which were closely correlated with stroke outcome [43]. However, we note that due to significant differences in the composition of human and rat microbiota, their direct comparison in terms of bacterial changes is complicated. For example, in one of the recent studies, a phylum-level analysis demonstrated a higher *Firmicutes*–*Bacteroidetes* ratio in human subjects, whereas rats showed a higher abundance of *Prevotella*; fecal levels of lactate were higher in mice and rats than human subject, and the highest level of acetate was observed in human feces [44]. Our findings indicated that the most abundant phyla in the healthy rats (before the TBI) were *Firmicutes* (66.1%) and *Bacteroidetes* (22.7%), which is consistent with the results of several studies of human subjects, whose gut microbiota includes *Firmicutes* and *Bacteroidetes* as major phyla.

Certainly, the composition of the intestinal microbiome of mammals is very different. In our study, we used a murine model; however, in spite of the differences in the intestinal microbiome of rats and humans, the study has a great translational potential. It resides in the data on post-TBI changes in certain microbial communities, primarily *Agathobacter*, which are consistent with clinical data [34,35]. Based on our findings, we were able to draw conclusions on the prospects of TBI therapy. First, as previously mentioned, fecal microbiota transplantation should gain ground as one of the most effective therapies for post-TBI dysbiosis [45]. Second, donor selection should be based on superior microbial diversity and balanced *Bacteroidetes–Firmicutes* ratio. This method has been proven effective in clinical trials on patients with recurrent *Clostridium difficile* infection [46].

Notably, we operated on the premise that biomedical models should be as close to human biology as possible. Hence, our study investigated the effect of the natural microbiome in its entirety that involves exposure to antigens. Given the significance of subject characteristics in microbiome studies [47], we opted for a path of “least interference” with the validity of the results to promote bench-to-bedside translation, i.e., we chose to model TBI in nonspecific-pathogen-free animals since the advantages of using SPF subjects are yet to be confirmed beyond a reasonable doubt: several researchers argue that in response to anesthesia and intervention, specific-pathogen-free animals display abnormal hemodynamic, hematological, and hemostatic phenotypes; their immune responses were atypical as well [48]. We deemed that SPF gut-microbiome heterogeneity [49] could be an impediment for achieving the goals of the present study.

Unexpectedly, there was also an increase in the number of lactobacilli in the post-TBI fecal samples. This finding is not consistent with the results obtained by other authors who showed a noticeable decrease in the abundance of these microorganisms following TBI [13,32] or spinal cord injuries [50]. Presumably, the discrepancy could be due to the time of sample collection. Depletion of lactobacilli is usually registered shortly after a CNS injury. We measured the abundance of *Lactobacillus* only seven days after the TBI; hence, we might have missed an initial decline in the level of lactobacilli immediately after the TBI. The increase observed one week after the CNS injury reflects some compensatory changes in the gut microbiome. It is also important to mention that some studies demonstrated a higher abundance of *Lactobacillus* species in patients with autism spectrum disorders [51], suggesting that the association between some CNS disorders and redundancy of beneficial commensal bacteria, such as *Lactobacillus*, might not be so straightforward.

Beta-diversity analysis demonstrated differences in the gut microbiota composition between pre-TBI and post-TBI samples, confirming the hypothesis that brain injury has a strong impact on the bacterial community of the gut. Notably, the bacterial genera that demonstrated significant changes after the TBI in rats are different from those observed in clinical studies of TBI patients. However, such studies in human subjects are rare and their findings are often contradictory. Thus, it is too early to draw final conclusions and compare results of clinical and experimental studies. Nevertheless, we can assume that acute brain injury (caused by TBI or ischemia) leads to significant gut dysbiosis, such as an increase in *Akkermansia* and decrease in *Bifidobacterium* abundance. TBI results in poorer microbial diversity, shifts in the abundances of some important representatives of the gut microbiota, and depletion of some beneficial commensal microorganisms, along with an overgrowth of less-beneficial bacteria or opportunistic pathogens. Such alterations, in turn, may have multiple effects on the host metabolism, which can be reflected in the metabolite profile of blood serum.

Hajiaghamemar M. et al. suggested that amino acid levels in blood serum were correlated with their content in brain tissue and can be prognostic markers of TBI [52]. Amorini A.M. et al. [53] found that in rats with severe TBI, citrulline levels in blood serum were lower than in those with mild TBI, but this result was insignificant: Citrulline levels significantly fluctuated after TBI, markedly increased during the first 6 h, decreased on day 2, and went back to normal on day 5. In our study, citrulline levels significantly dropped on days 3 and 7 after the TBI; however, the decrease in Group 2 (showing negative neurological dynamics) was insignificant. Even though we did not find significant differences in citrulline levels between the two groups, we cannot dismiss citrulline level in blood serum as a prognostic factor for TBI outcomes and severity. Our study showed that TBI affects citrulline level by decreasing it in the acute period. The mechanism behind this process is not clear. It is known that L-citrulline has a cerebrovascular protective effect: it was associated with the expression of endothelial nitric oxide synthase and prevention of neural cell death [54]. Some researchers found that the level of citrulline in peripheral blood can be used as a marker of GI dysfunction and injury [55,56]. TBI triggers gut injury and can cause intestinal mucosal atrophy, damage to the intestinal barrier and inflammation [57]. From this point of view, citrulline in blood serum can reflect TBI-induced GI dysfunction. This assertion can be proved by the observed changes in the taxonomic composition and diversity of the post-TBI gut microbiota samples.

TBI can cause serum proline levels to drop [58]: Proline levels are associated with brain damage severity [58] and can be a biomarker of post-traumatic neurological deficit [59]. In our study, an increase in proline levels in the post-TBI blood samples was not significant, but its pre-TBI levels in rats with no positive neurological dynamics (Group 2) were significantly higher than in Group 1. Increased levels of proline were not protective against the injury and probably could be a predictor of negative neurological outcomes in rats with TBI. In several studies, inhibition of proline metabolism is proposed as a therapeutic approach for a variety of diseases [60,61]. However, the role of proline in brain-injury pathogenesis can be more complicated and its therapeutic value can be not so obvious. Some studies reported a possible anti-inflammatory effect of proline in the brain cortex [62]; others, however, demonstrated a toxic effect of hyperprolinemia [63]. The results of our study suggest that increased proline levels in blood serum can predict poor post-TBI recovery prospects.

In our study, serum tryptophan levels decreased on day 3 after the TBI and returned to normal on day 7. This finding is supported by Zhang Z. et al. who reported a decrease in tryptophan and serotonin ratios and an increase in interleukins in rabbit brain on days 7–21 after TBI [64]; however, the normalization of tryptophan levels on day 7 requires a more in-depth study. However, we did not observe any significant changes in the two groups. Serum tryptophan levels can also change as the result of its metabolism in kidneys and may drop due to an acute kidney injury [65,66]; however, renal failure after TBI has not been previously reported. Tryptophan levels were not associated with recovery after the TBI, but their fluctuation can be associated with the brain–gut-microbiome axis. Changes in the post-TBI samples, such as an insignificantly decreased abundance of *Bacteroidaceae* and increased abundance of *Firmicutes*, possibly confirm this assertion. Some authors report an association between postsurgical GI dysfunction, followed by neuropsychiatric disorders and impaired tryptophan metabolism [64], a and decrease in *Bacteroidaceae* and increase in *Firmicutes* bacteria with a concurrent decrease in *Proteobacteria* [67].

Serum levels of free carnitine insignificantly decreased on day 3 after the TBI; however, the difference between carnitine levels on day 3 compared to their increase close to the pre-TBI levels on day 7 was significant. Fluctuations in carnitine ratios were significant. This observation is consistent with the results obtained by Vardon Bounes F. et al. who reported decreased levels of free carnitine after severe TBI [67]. It should be noted, however, that the fluctuations in our study were not as evident, which could be attributed to a milder TBI induced in our study. There were no differences between the two groups in this respect; serum levels of carnitine were not of any diagnostic value for recovery rate after the TBI in the acute period. carnitine and its metabolism by the GI microbiota are frequently considered to have a cardioprotective effect [68]. Moreover, severe TBI is known to be associated with cardiac disorders, which can affect TBI outcomes [69]. Changes in serum carnitine levels after TBI can affect the functioning of the GBA and have a prognostic value for recovery after severe TBI.

## 5. Conclusions

The brain–gut-microbiome axis is a tightly organized system: any external impact on one of its components echoes in every other, producing a complex, multifaceted effect. Therefore, TBI not only affects the brain, but alters the whole system along the axis, specifically the gut microbiome, a powerful regulator of health and wellbeing. Once the composition of the gut microbiome has been changed by TBI, a wide variety of shifts occur that disturb multiple organ systems. The changed gut microbiome, in its turn, affects the severity of TBI and neurological recovery prospects. The TBI–microbiome interaction is reflected in measurable biochemical and metabolic changes. These changes could be a valuable prognostic tool for post-TBI neurological recovery and could serve as a basis for developing therapeutic approaches to minimizing TBI impact and outcomes.

## Figures and Tables

**Figure 1 cells-11-01409-f001:**
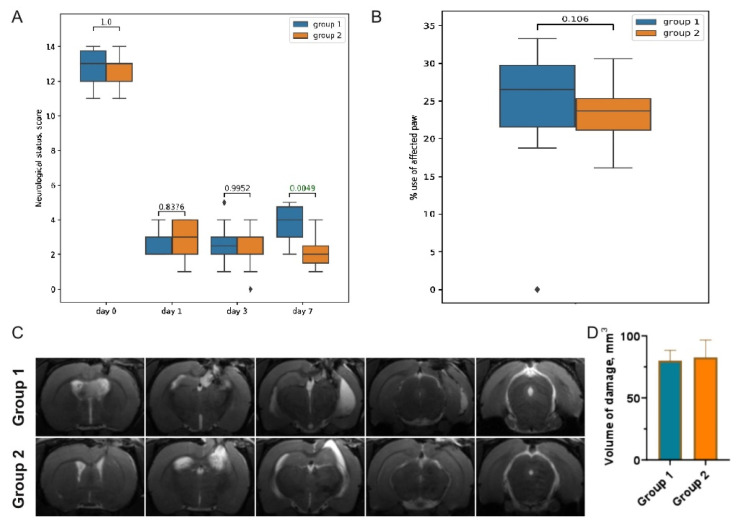
(**A**) LPT scores before (day 0) and on days 1, 3, and 7 after the TBI. Grouping was based on the observed dynamics of neurological recovery. (**B**) TBI-induced forelimb asymmetry measured with the Cylinder Test on day 7 after the TBI. (**C**) Representative T2-weighted MRIs of the coronal brain sections (0.8 mm thick, from rostral (left) towards caudal (right)), taken on day 7 after the TBI. (**D**) Morphometric analysis of the extent of the brain damage evaluated based on T2-weighted images.

**Figure 2 cells-11-01409-f002:**
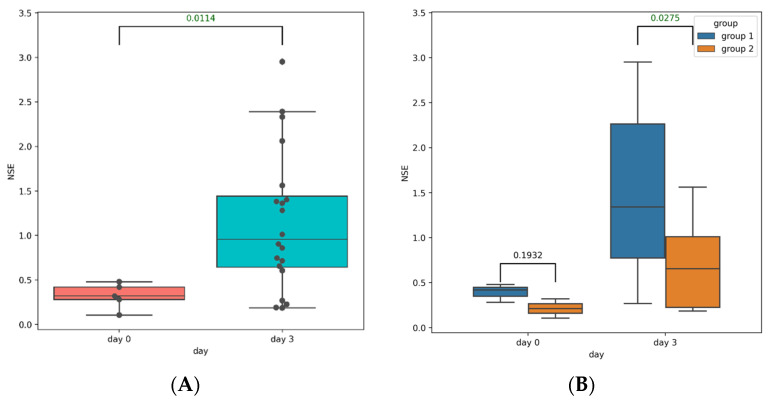
(**A**) NSE levels in the pre-TBI (blue box) and post-TBI (orange box) serum samples of all rats. (**B**) NSE levels in the pre-TBI (left) and post-TBI (right) serum samples in Group 1 (blue box) and Group 2 (orange box).

**Figure 3 cells-11-01409-f003:**
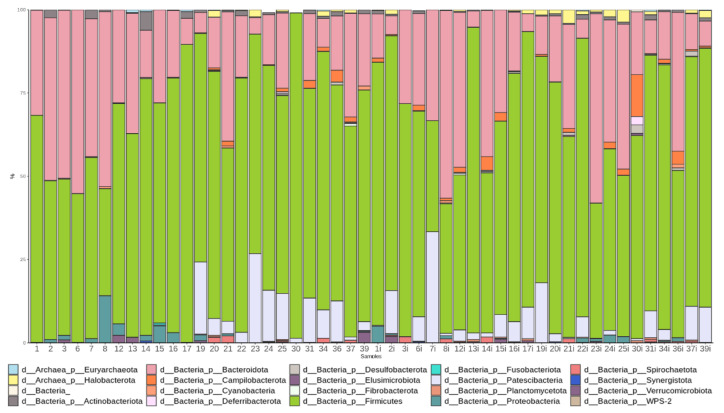
Abundance of the main bacterial phyla before (samples 1 through 25) and after the TBI (samples 1i through 25i).

**Figure 4 cells-11-01409-f004:**
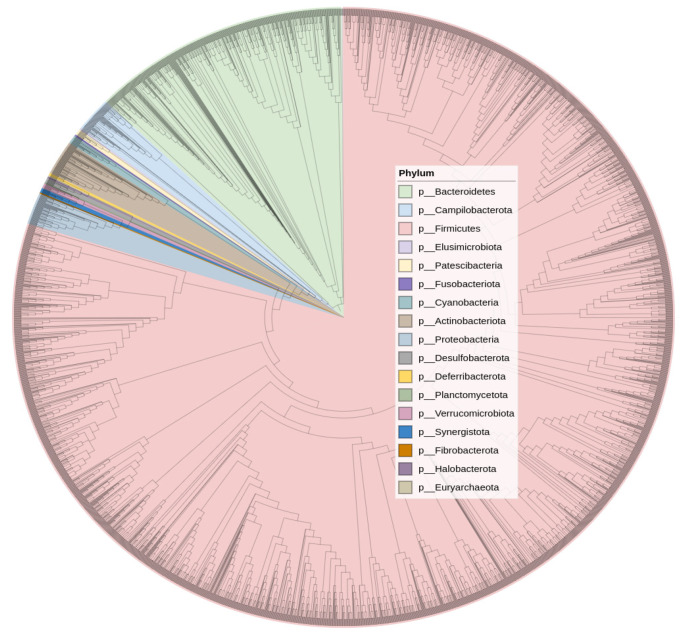
Phylogenetic tree diagram.

**Figure 5 cells-11-01409-f005:**
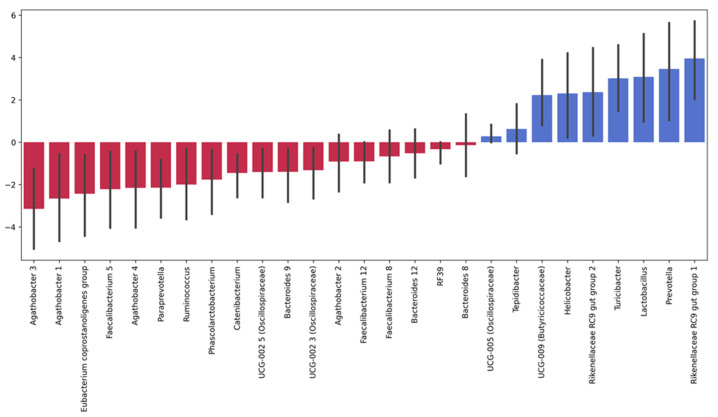
Significant post-TBI changes in the relative abundance of intestinal microorganisms. Note: The height of each bar depicts a fold change in the abundance of a specific microorganism: blue bars show increase and red bars show decrease in post-TBI abundance. Error bars denote CI95% for each change. We calculated the fold change in each microorganism as the mean of its fold changes across all samples before and after the TBI.

**Figure 6 cells-11-01409-f006:**
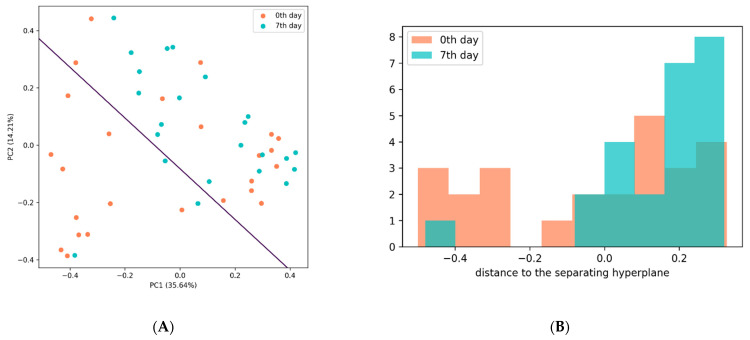
(**A**) Support-vector-machine analysis of the discrimination between the pre-and post-TBI samples based on PC1 and PC2. (**B**) Distribution of the distances to the separating hyperplane in the pre-and post-TBI samples.

**Figure 7 cells-11-01409-f007:**
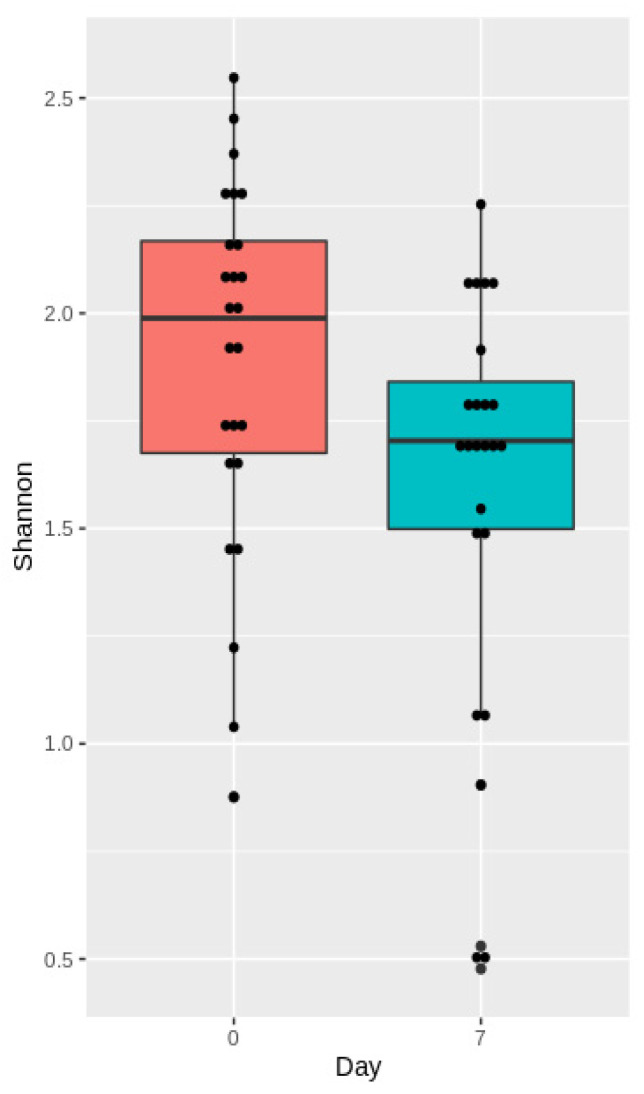
Shannon indices before (day 0) and after the TBI (day 7).

**Figure 8 cells-11-01409-f008:**
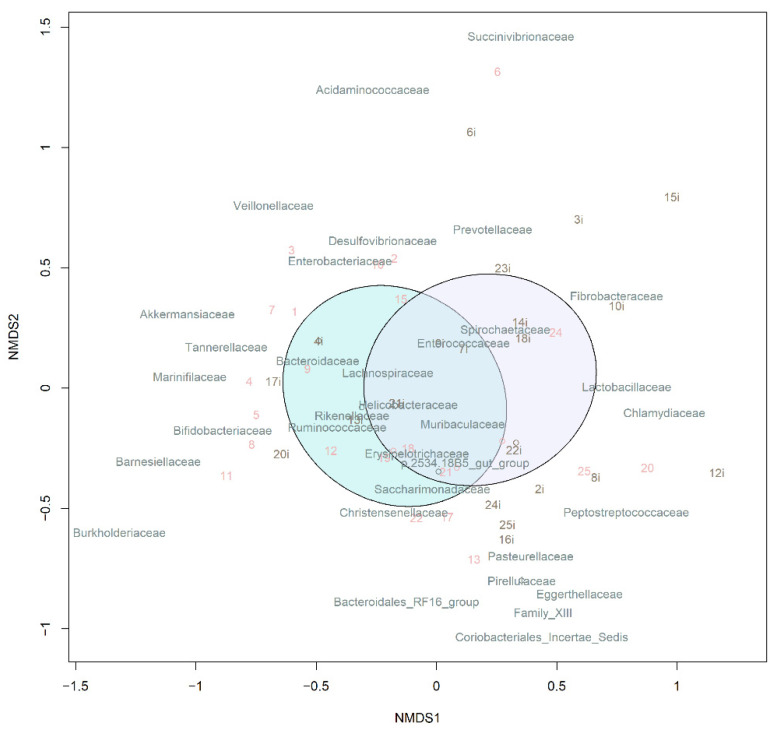
NMDS of beta-diversity in the samples. Ellipsoids correspond to 95% confidence intervals. Note: The purple and blue ellipsoids show the microbiome before and after the TBI, respectively.

**Figure 9 cells-11-01409-f009:**
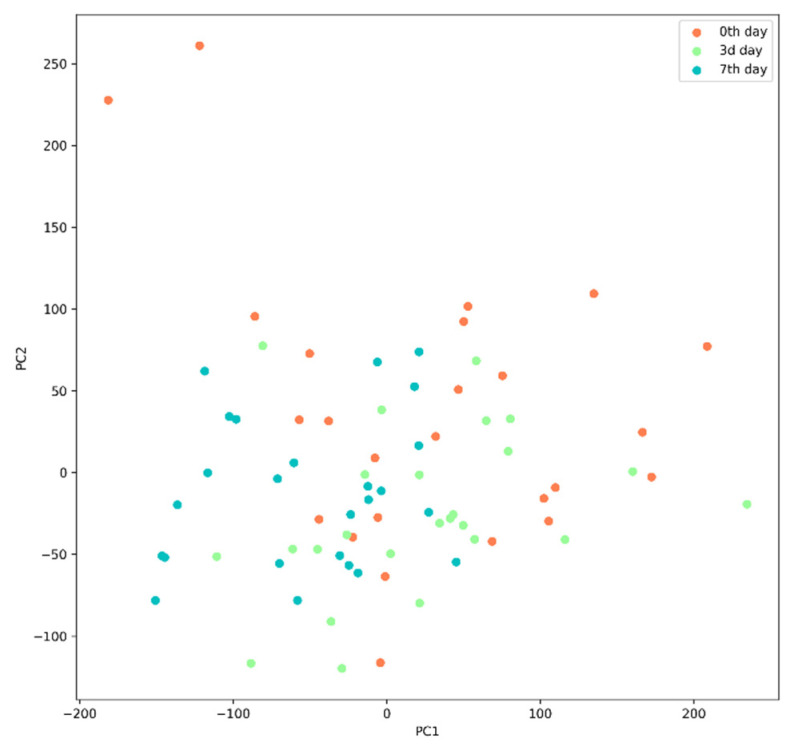
Amino acid levels in blood serum before and on days 3 and 7 after the TBI.

**Figure 10 cells-11-01409-f010:**
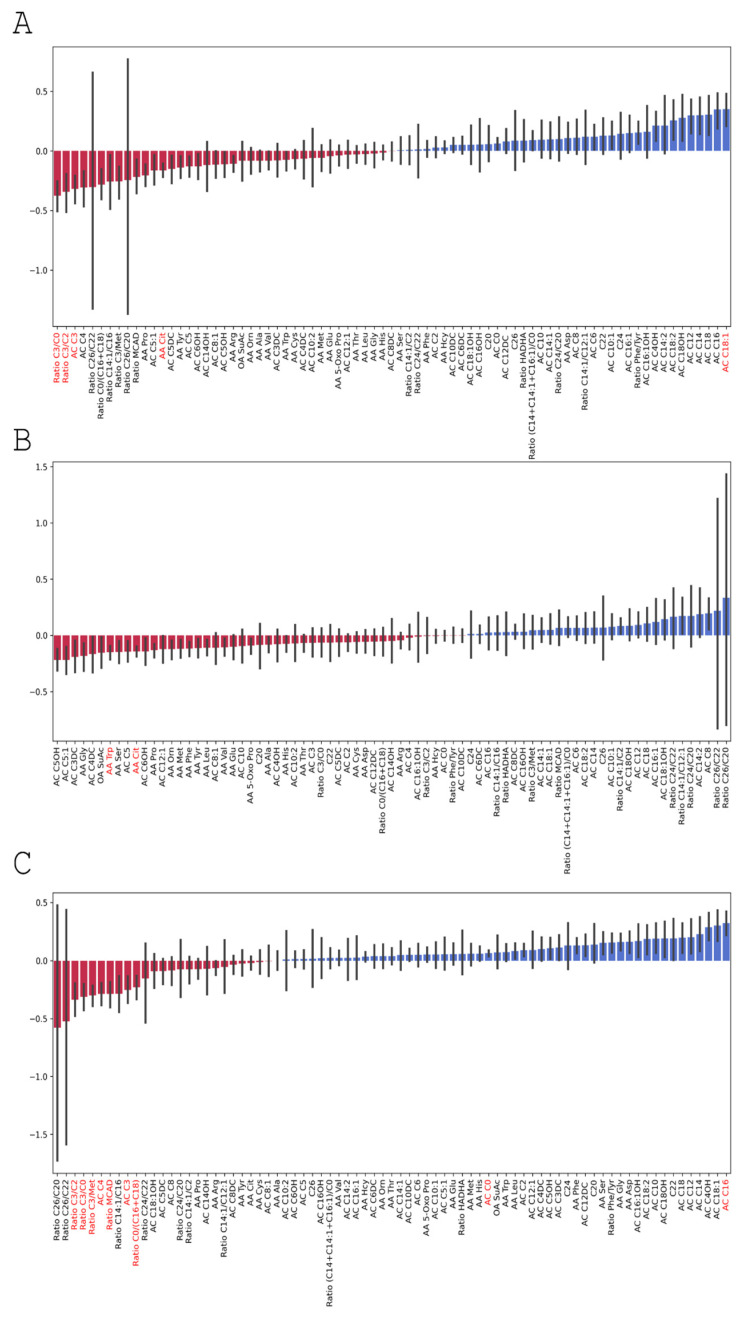
Fold changes in amino acid levels (**A**) before and on day 7 after the TBI; (**B**) before and on day 3 after the TBI; (**C**) on days 3 and 7 after the TBI. Significant results are shown in red.

**Figure 11 cells-11-01409-f011:**
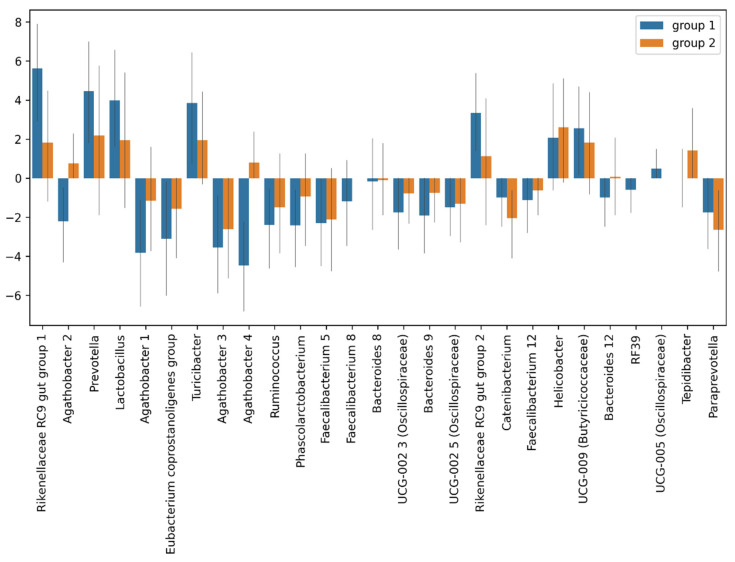
Fold changes in the relative abundance of the post-TBI gut microbiota species in Group 1 (blue) and Group 2 (red).

**Figure 12 cells-11-01409-f012:**
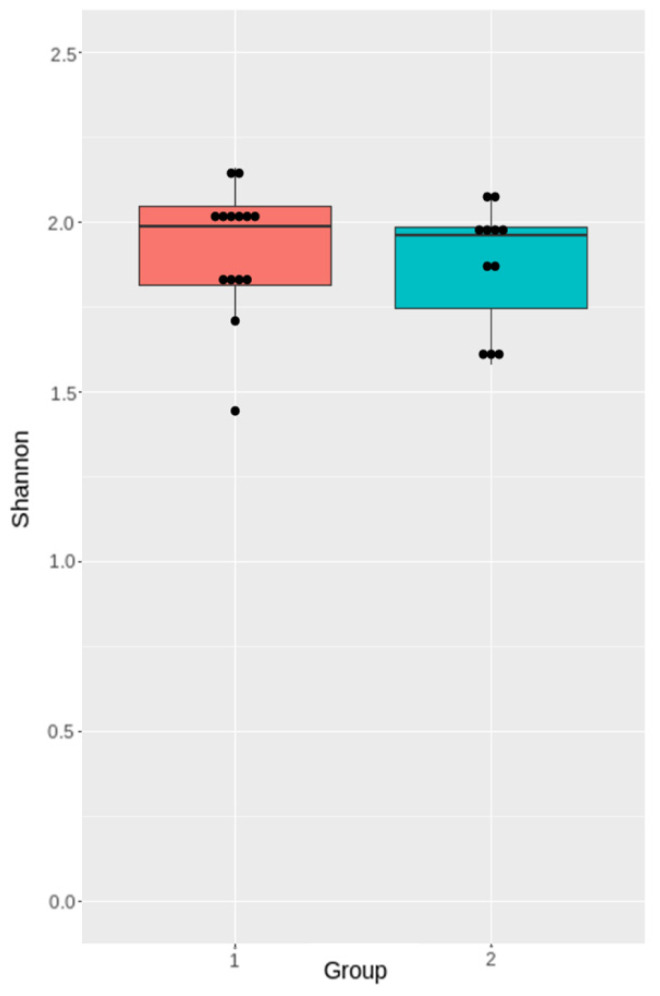
Shannon diversity indices of the pre-TBI gut microbiota in Group 1 and Group 2.

**Figure 13 cells-11-01409-f013:**
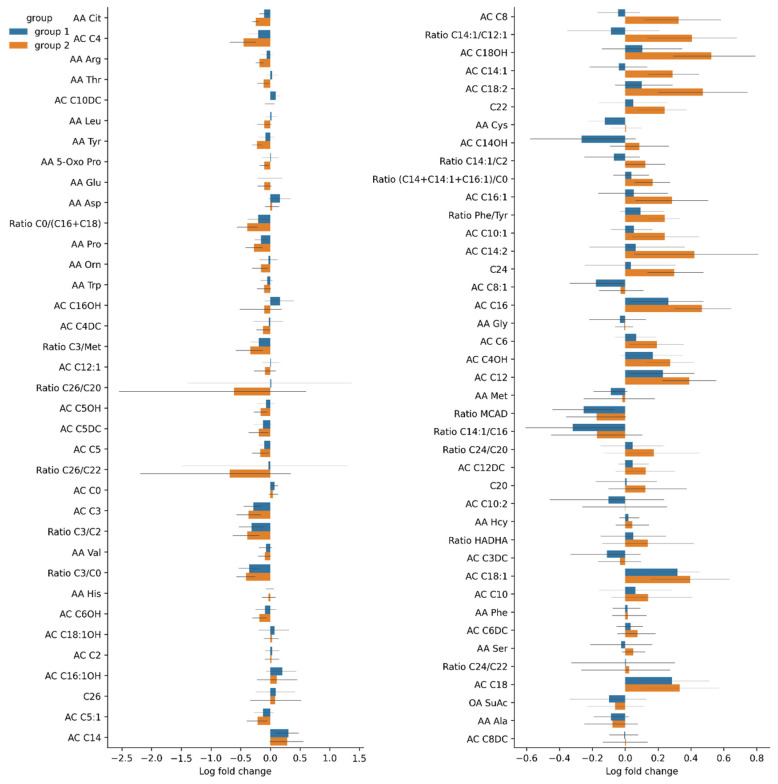
Fold changes in pre-TBI (day 0) amino acid levels in Groups 1 and 2. Significant results are presented in red.

**Figure 14 cells-11-01409-f014:**
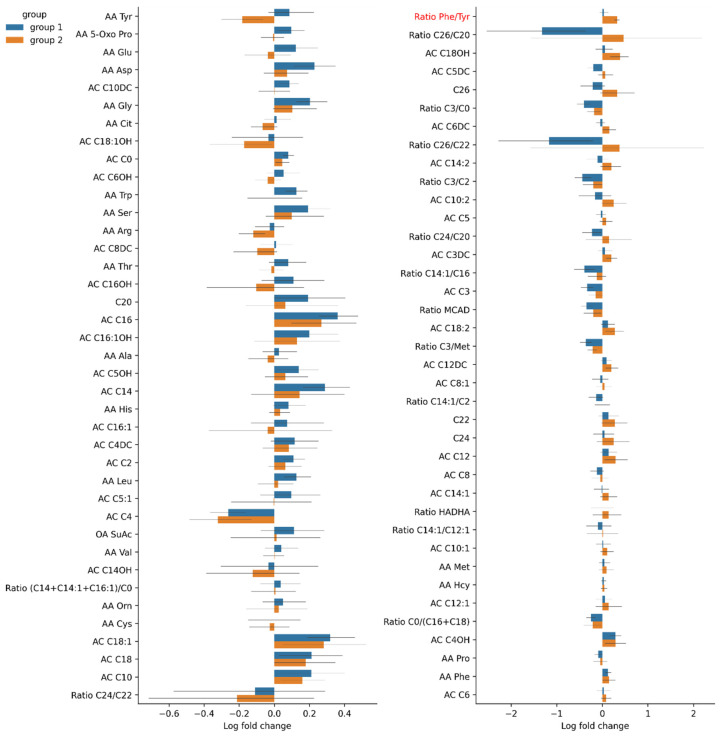
Fold changes in post-TBI amino acid levels in Groups 1 and 2 between days 3 and 7. Significant results are presented in red.

**Table 1 cells-11-01409-t001:** Significant changes in amino acid levels in blood serum on days 3 and 7 after the TBI, compared to the baseline. Note: Significant changes are marked with an asterisk.

Amino Acid	0 Day—Before the Brain Injury	3rd Day	7th day	*p* ^0–7^	*p* ^3–7^	*p* ^0–3^
Citrulline. AA Cit	37.63 (6.303)	32.65 (5.788)	31.99 (3.532)	$4.41\times 10^{-4}$ *	$1.23\times 10^{-1}$	$6.33\times 10^{-5}$ *
Tryptophan. AA Trp	4.76 (1.410)	4.10 (1.057)	4.37 (1.090)	$5.20\times 10^{-2}$	$8.14\times 10^{-2}$	$3.55\times 10^{-4}$ *
Free carnitine. AC C0	16.61 (4.049)	16.48 (4.860)	17.51 (3.971)	$2.40\times 10^{-2}$	$3.96\times 10^{-4}$ *	$9.77\times 10^{-1}$
Palmitoylcarnitine.AC C16	0.04 (0.011)	0.04 (0.012)	0.05 (0.016)	$1.18\times 10^{-3}$	$1.05\times 10^{-4}$ *	$3.46\times 10^{-1}$
Oleoylcarnitine. AC C18:1	0.02 (0.008)	0.02 (0.011)	0.033 (0.009)	$2.27\times 10^{-4}$ *	$7.46\times 10^{-4}$	$4.65\times 10^{-1}$
Propionyl carnitine. AC C3	0.81 (0.494)	0.74 (0.361)	0.59 (0.394)	$2.28\times 10^{-4}$ *	$6.07\times 10^{-7}$ *	$2.78\times 10^{12}$
Butyryl carnitine. AC C4	0.25 (0.086)	0.24 (0.079)	0.18 (0.092)	$8.20\times 10^{-4}$	$1.62\times 10^{-4}$ *	$3.76\times 10^{-1}$
Ratio free carnitine/(palmitoylcarnitine + stearoylcarnitine). Ratio C0/(C16 + C18)	349.79 (120.934)	331.71 (128.36)	263.67 (93.066)	$1.67\times 10^{-3}$	$6.07\times 10^{-4}$ *	$3.04\times 10^{-1}$
Ratio propionyl carnitine/free carnitine. Ratio C3/C0	0.05 (0.027)	0.05 (0.026)	0.034 (0.020)	$2.63\times 10^{-5}$ *	$1.28\times 10^{-4}$ *	$2.60\times 10^{-1}$
Ratio propionyl carnitine/acetylcarnitine. Ratio C3/C2	0.07 (0.037)	0.07 (0.042)	0.05 (0.038)	$3.33\times 10^{-4}$ *	$5.26\times 10^{-4}$ *	$8.86\times 10^{-1}$
Ratio propionyl carnitine/methionine. Ratio C3/Met	0.026 (0.020)	0.03 (0.013)	0.02 (0.011)	$1.10\times 10^{-3}$	$9.03\times 10^{-5}$ *	$4.29\times 10^{-1}$
Ratio (hexanoylcarnitine + octanoalkarnitine + decenoalkarnitine)/(palmitoylcarnitine + stearoylcarnitine + oleoylcarnitine). Ratio MCAD	0.65 (0.239)	0.68 (0.187)	0.51 (0.160)	$6.09\times 10^{-3}$	$1.82\times 10^{-4}$ *	$6.27\times 10^{-1}$

**Table 2 cells-11-01409-t002:** Significant changes in amino acid levels in blood serum on days 3 and 7 after the TBI compared to the baseline. Note: Significant changes are marked with an asterisk.

Amino Acid	Group 1			Group 2			*p*-Value for Pre-TBI	*p*-Value for Fold Change (Day 3 vs. 7)
	Pre-TBI	Day 3	Day 7	Pre-TBI	Day 3	Day 7		
Proline. AA Pro	79.25 (8.379)	74.34 (13.034)	68.56 (13.783)	101.55 (13.671)	81.27 (19.061)	78.62 (22.212)	$7.94\times 10^{-5}$ *	0.44
Ratio phenylalanine/tyrosine. Ratio phe/tyr	0.89 (0.161)	0.93 (0.152)	0.97 (0.172)	0.87 (0.218)	0.80 (0.113)	1.10 (0.143)	$1.39\times 10^{-2}$	$2.47\times 10^{-4}$ *

## Data Availability

The data generated during and/or analyzed during the present study are available from the corresponding author upon reasonable request.

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
