# Peer review of "Effects of Traumatic Brain Injury on the Gut Microbiota Composition and Serum Amino Acid Profile in Rats"

_cells, 2022, doi:10.3390/cells11091409_

Round 1

Reviewer 1 Report

I thank the reviewers' response which has been well elaborate and high;y systematic.

I only disagree about the Specific pathogen-free (SPF) animals; I leave it to the editor's decision. Alternatively,,  the authors can have this in the discussion

Overall, they have adequately answered my concerns,

good work

Author Response

Response

Thank you very much for your positive report. We appreciate your critical review, valuable comments, and indispensable contribution to improving our manuscript. We understand you concerns about SPF animals and address them in the Discussion section:

“Notably, we operated on the premise that biomedical models should be as close to human biology as possible. Hence, our study investigated the effect of the natural microbiome in its entirety that involves exposure to antigens. Given the significance of subject characteristics in microbiome studies [77], we opted for a path of “least interference” with the validity of the results to promote bench-to-bedside translation, i.e., we chose to model TBI in non-specific-pathogen-free animals since the advantages of using SPF animals are yet to be confirmed beyond a reasonable doubt: several researchers argue that, in response to anesthesia and intervention, specific-pathogen-free animals display abnormal hemodynamic, hematological and hemostatic phenotypes; their immune responses were atypical as well [78]. We deemed that SPF gut microbiome heterogeneity [79] could be an impediment for achieving the goals of the present study.” 

Reviewer 2 Report

  1. All the figures should be improved for clear demonstration. Some of them should be combined.
  2. In figure 2A, why did the author compare the NSE level of Days 3 of Group 2 with Day 0 of Group 1?
  3. What’s the relationship between Agathobacter species and metabolites in serum? It should be explained in detail. Is there any difference between Group 1 and 2?
  4. What’s the translational value of this paper? Has any findings been validated in clinic study?
  5. The language should be edited by native speakers.

Round 2

Reviewer 2 Report

The manuscript has been improved in this version. It has answered my concerns. It could be accepted after carefully revison of grammar and typos.

This manuscript is a resubmission of an earlier submission. The following is a list of the peer review reports and author responses from that submission.

Round 1

Reviewer 1 Report

In this manuscript, Darya Lisovaya found TBI caused significant changes in microbiota and related metabolites from rat model. Several points should be addressed.

  1. In Line 279 Page 6, “The most abundant phyla in the pre-TBI samples were Firmicutes (66.1%) and Bac teroidetes (22.7%). On day 7 after the TBI, the proportion of Firmicutes decreased to 68.0% and Bacteroidetes to 19.8% (Figure 2)”. However, 68.0% is more than 66.1%.
  2. The section of Materials and Methods should be improved. Some experiments such as limb-placing test can be described in detail. In contrast, only key points of PCR procedure should be demonstrated.
  3. How does the author divide the mice into group 1 and group 2? Detailed criteria should be mentioned in the manuscript.
  4. Microorganisms with significant dysregulation should be labeled properly in Fig. 4 and Fig 11.
  5. Do Agathobacter species show different abundance in group 1 and group 2?
  6. How does the findings of microorganisms and related metabolites correlate with clinic observations in TBI patients? Discussions should be carried out in detail.
  7. Are the metabolites such as citrulline or tryptophan derived from the microorganisms dysregulated in the rat model?
  8. They compared the microbiota of pre-TBI and 7-day post TBI rats. Why does the author chose this time point? At this time point, several rats were recovered. Does it interfere with the correct conclusion?

Reviewer 2 Report

the work by Lisovaya et al assessed altered metabolomics and gut microbiome post-TBI. there are major weaknesses in the work.

It lacks any mechanistic and validation in conducting this work.

1-the work should assess the gut permeability as part of the altered microbiome, what they are reporting is only descriptive and has no validation.

2- the TBI procedure performed should be fully described and it seems nonconevtional and Immunoflorescnce of the brain should be performed.

3- the work lacks controls and shams which render the results as not conclusive

4- the work relied on neurological scoring as a discriminant of the two groups n=11 vs n=14, behavioral testing should have been performed to conduct TBI validity.

5-the work has no injury marker assessment of the brain

6- you cant perform any microbiome without having a germ free facility and apparently this is not present

7-biomarkers of TBI should have been assessed